# The oncogenic fusion protein DNAJB1-PRKACA can be specifically targeted by peptide-based immunotherapy in fibrola-mellar hepatocellular carcinoma

The DNAJB1-PRKACA fusion transcript is the oncogenic driver in fibrolamellar hepatocellular carcinoma, a lethal disease lacking specific therapies. This study reports on the identification, characterization, and immunotherapeutic application of HLA-presented neoantigens specific for the DNAJB1-PRKACA fusion transcript in fibrolamellar hepatocellular carcinoma. DNAJB1-PRKACA-derived HLA class I and HLA class II ligands induce multifunctional cytotoxic CD8[+] and T-helper 1 CD4[+] T cells, and their cellular processing and presentation in DNAJB1-PRKACA expressing tumor cells is demonstrated by mass spectrometry-based immunopeptidome analysis. Single-cell RNA sequencing further identifies multiple T cell receptors from DNAJB1-PRKACA-specific T cells. Vaccination of a fibrolamellar hepatocellular carcinoma patient, suffering from recurrent short interval disease relapses, with DNAJB1-PRKACA-derived peptides under continued Poly (ADP-ribose) polymerase inhibitor therapy induces multifunctional CD4[+] T cells, with an activated T-helper 1 phenotype and high T cell receptor clonality. Vaccine-induced DNAJB1-PRKACA-specific T cell responses persist over time and, in contrast to various previous treatments, are accompanied by durable relapse free survival of the patient for more than 21 months post vaccination. Our preclinical and clinical findings identify the DNAJB1-PRKACA protein as source for immunogenic neoepitopes and corresponding T cell receptors and provide efficacy in a single-patient study of T cell-based immunotherapy specifically targeting this oncogenic fusion.

T cell-based immunotherapies, comprising immune checkpoint inhibitors (ICIs), CAR-T cells, and bispecific T cell engager antibodies achieved a breakthrough in the treatment of malignant disease[1–5], adoptive T cell transfer and vaccination strategies hold further promise[6–10]. However, these therapies, which rely on the rejection of cancer cells through recognition of tumor antigens and T cell-mediated cytotoxicity are still only available and effective in small subsets of cancer patients and single tumor entities. One main problem

for the development of antigen-specific immunotherapies is the lack of suitable target structures that show natural, highly frequent, and tumor-exclusive presentation on the cell surface of tumor cells and are recognized by the immune system[11]. Tumor antigens are represented by HLA-independent surface molecules or by HLA class I- and HLA class II-presented T cell epitopes, originating from intracellular proteins[12]. In terms of the latter, neoepitopes arising from tumor-specific mutations have been identified in recent years as the main specificity of

e-mail: juliane.walz@med.uni-tuebingen.de

anti-cancer T cell responses induced by ICIs, and were in turn suggested as prime candidates for T cell-based immunotherapy approaches[13–15]. In line, response to ICIs correlates with high tumor somatic mutational burden, and neoepitope-based immunotherapies have been applied in individual tumor patients[16–18]. However, patient/tumor-specificity and intratumoral heterogeneity of somatic mutations, as well as the limited number of somatic mutations that are ultimately translated, processed, and presented as HLA-restricted neoepitopes on the tumor cells[8,13,19–21] restrict the broad applicability of these antigens in particular in low-mutational burden cancer entities[22]. Recently, fusion transcripts, whose products often represent clonal oncogenic drivers, were identified as the source of highly immunogenic neoepitopes, and T cell responses against such fusion protein-derived neoepitopes were detected in patients receiving ICIs and correlated with treatment response[23].

The DNAJB1-PRKACA fusion transcript links exon 1 of the DnaJ homolog subfamily B member 1 gene (DNAJB1) to exon 2–10 of the cAMP-dependent protein kinase catalytic subunit alpha gene (PRKACA)[24]. In FL-HCC, the DNAJB1-PRKACA fusion transcript is detectable in 100% of patients and has been identified as the oncogenic driver in tumor pathogenesis[25,26] indicating expression of the fusion transcript in all tumor cells. FL-HCC is a devastating tumor disease with a 5-year survival of only 45%, which typically affects children and young adults with no history of primary liver disease. The frequency of FL-HCC diagnosis is continuously increasing to ~5% of all liver cancers today[27,28]. To date, surgical resection is the only effective therapy if the cancer is diagnosed before the occurrence of metastases, and long-term survival is jeopardized by tumor recurrence calling for the development of specific treatment options for FL-HCC patients[26,29]. The oncogenic fusion protein DNAJB1-PRKACA represents an attractive target for the development of novel therapies for this devastating tumor disease. Moreover, the recent identification of other cancer entities that express the DNAJB1-PRKACA fusion transcript gives the prospect that targeting the fusion protein might improve treatment options in multiple cancer entities[30].

Here, we show that the DNAJB1-PRKACA fusion transcript is a prime source for broadly applicable neoepitopes and provide the first evidence for their efficacy in immunotherapy approaches in an FL-HCC patient.

## Results
### The DNAJB1-PRKACA fusion gene is a source of HLA class I and HLA class II neoantigens and induces DNAJB1-PRKACA-specific CD4+ and CD8+ T cells
To identify potential DNAJB1-PRKACA fusion gene-specific HLA ligands an in silico prediction was conducted based on the DNAJB1-PRKACA protein sequence (NCBI accession 4WB7_A). The in silico prediction workflow using the algorithm NetMHCIIpan identified nine unique binding cores of nine amino acid (AA) lengths for a total of 1290 different HLA class II alleles within the 24 AA fusion region of the DNAJB1-PRKACA protein (Fig. 1a, b). 83.5% of these alleles represent HLA-DP combinations, 11.6% and 5.0% are HLA-DQ or HLA-DR combinations, respectively. Focusing on the population frequencies of the top four alleles covered by the fusion region (HLA-DPA1*02:02-DPB1*05:01, HLA-DPA1*01:03-DPB1*05:01, HLA-DPA1*02:01-DPB1*05:01, and HLA-DPA1*01:03-DPB1*09:01) in a publicly available HLA-DP, -DQ, and -DR allele typed population (Japan pop 17; http://www.allelefrequencies.net), 41.4%, 39.1%, 17.9%, and 12.5% of the donors were calculated to carry the respective allotype, suggesting very broad applicability of this fusion neoepitope based on its' predicted promiscuous HLA binding. The core sequence RYGEEVKEF, located directly in the middle of the fusion transcript (5 AAs on exon 1 (DNAJB1), 4 AAs on exon 2 (PRKACA)) is predicted to bind to the majority of different HLA class II alleles (60.52% of possible allele/binding core combinations). For HLA class I, 13 DNAJB1-PRKACA-derived HLA ligands were identified for the

20 most frequent HLA class I allotypes of the European population, using a prediction workflow combining the algorithms SYFPEITHI and NetMHCpan (Fig. 1c, Table 1). These 13 HLA class I allotypes within the 22 AA peptide KREIFDRYGEEVKEFLAKAKED (P$_{II-1}$), spanning the fusion region of the DNAJB1-PRKACA protein, cover 96.6% and 93.8% of the European and world population with at least one HLA allotype, respectively (Fig. 1d, Supplementary Fig. S1a). Of note, the HLA class II-binding core RYGEEVKEF was also predicted as HLA class I ligand binding the alleles HLA-A*24:02, -C*04:01, -C*06:02, and -C*07:02.

Cellular presentation of DNAJB1-PRKACA-derived HLA-presented peptides was shown by liquid chromatography–coupled tandem mass spectrometry (LC–MS/MS) of differentiated and matured monocyte-derived dendritic cells (moDCs; Supplementary Fig. S1b) from healthy volunteers (HV) loaded with the isotope-labeled 22 AA peptide P$_{II-1}$. The mass spectrometric (MS) identified fragment ion spectra of the experimentally eluted P$_{II-1}$ were validated using the synthetic peptide (Supplementary Fig. S1c). P$_{II-1}$ and 12 shorter length variants were identified by MS and predicted to bind to the HLA-DP allele DPA1*01:03-DPB1*05:01 of the respective HV6 with the best NetMHCIIpan binding rank of 0.74 for the binding core RYGEEVKEF (Fig. 1b, Table 2, Supplementary Table 1). De novo priming of CD4+ T cells from HVs with P$_{II-1}$-loaded mature moDCs induced multifunctional P$_{II-1}$-specific CD4+ T cells, which showed expression of CD107a and CD154 as well as production of interleukin-2 (IL-2), interferon-γ (IFN-γ), and tumor necrosis factor (TNF) upon P$_{II-1}$ stimulation (Fig. 1e). Refolding of the DNAJB1-PRKACA protein fusion-derived ligands RYGEEVKEF (P$_{A*24}$, SYFPEITHI score of 74.19% and NetMHCpan rank of 0.018), and EIFDRYGEEV (P$_{A*68/A*02}$, A*68:02 NetMHCpan rank of 0.106) was conducted to an HLA-A*24:02-P$_{A*24}$ monomer and HLA-A*68:02-P$_{A*68/A*02}$ monomer, respectively. These monomers were used to build artificial antigen-presenting cells (aAPCs) for aAPC-based priming of CD8+ T cells of HVs to validate their immunogenicity. aAPC-based priming induced P$_{A*24}$-specific and P$_{A*68/A*02}$-specific CD8+ T cells with frequencies of up to 15.7% (median 4.1%) and 1.1% (median 0.7%) peptide-specific T cells, respectively (Fig. 1f, g, Supplementary Fig. S1d, e, Table 1). Flow cytometry-based functional characterization of P$_{A*24}$-specific and P$_{A*68/A*02}$-specific CD8+ T cells showed a polyfunctional phenotype reflected by IFN-γ, TNF, and CD107a production/expression (Fig. 1h, i).

### In-depth characterization of DNAJB1-PRKACA fusion protein-specific CD8+ T cells and single-cell TCR sequencing
We further conducted an in-depth characterization of P$_{A*24}$-specific CD8+ T cells using single-cell RNA-sequencing analysis of flow cytometry-based bulk sorted P$_{A*24}$-specific CD8+ T cells of HV 1 and HV2 which showed a high expression of cytotoxicity markers comprising amongst others GNLY, GZMA, GZMB, GZMK, PRF1, and NKG7, paired with a low expression of the exhaustion marker PDCD1 (Fig. 2a). P$_{A*24}$-specific CD8+ T cells specifically lysed P$_{A*24}$-loaded autologous CD8− cells in vitro with up to 82.4% lysis of target cells in comparison to unspecific effector cells at various effector to target ratios (Fig. 2b). Single-cell T cell receptor (TCR) sequencing of the P$_{A*24}$-specific CD8+ T cell bulks from the two HVs showed high clonality of one dominant TCR clone for HV2 and two clones for HV1 (Supplementary Fig. S2a, b and Supplementary Table 2). There was no overlap of the V(D)J genes of the three TCR clones, but the complementary determining region (CDR) 3-β sequences of the main clone from HV1 and HV2 shared a high sequence identity and similarity of 69.2% and 76.9%, respectively (Fig. 2c, Supplementary Fig. S2c). The CDR3-α sequence of the main TCR clone of HV1 showed, in terms of physiochemical properties of the AA sequences, opposing characteristics regarding chemical groups in comparison to the target peptide P$_{A*24}$, which could not be observed in the CDR3-α motif cluster of a negative control dataset of published TCR sequences (Supplementary Fig. S2d).

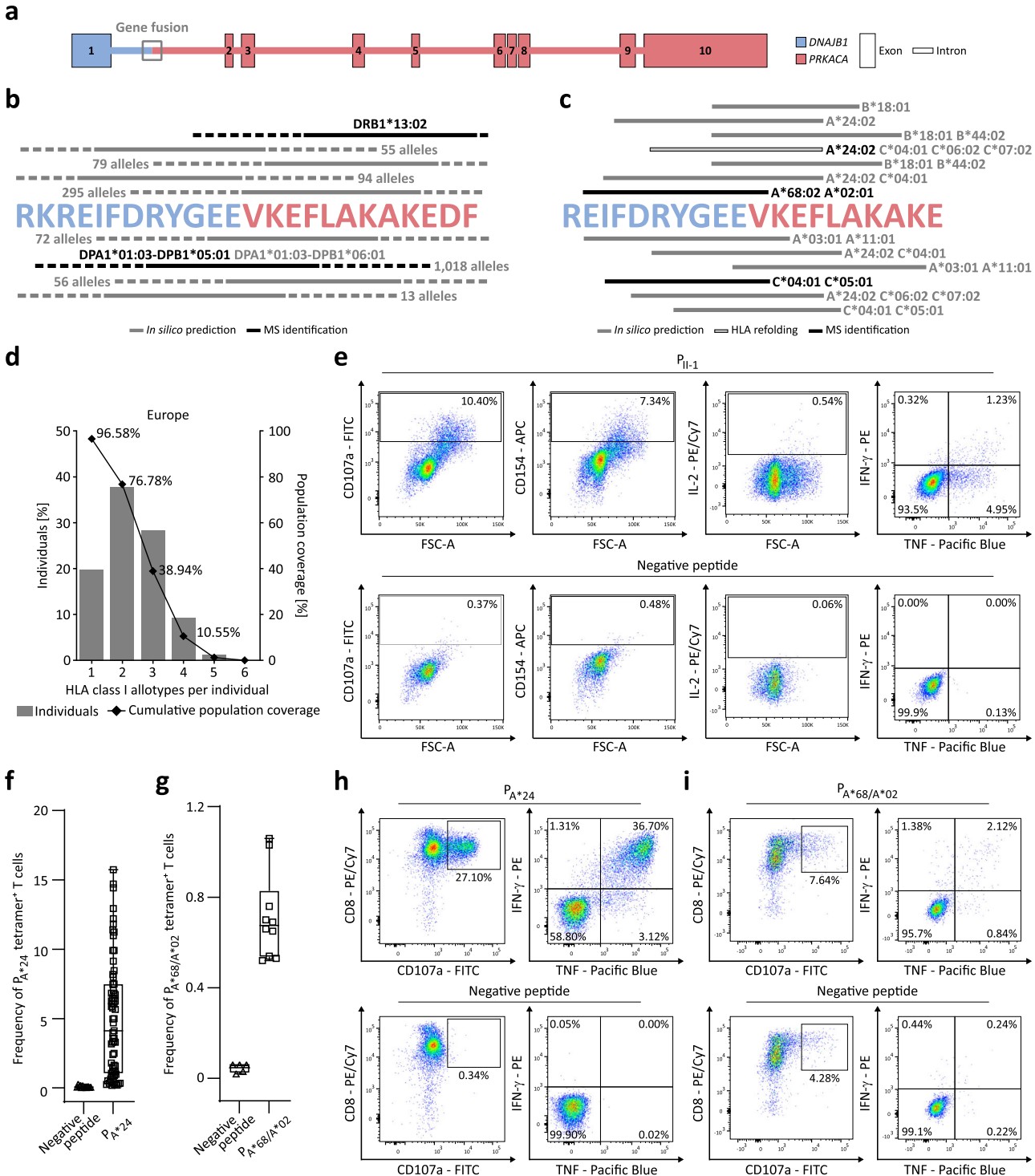

## Doxycycline-inducible DNAJB1-PRKACA fusion gene expression leads to cellular processing and presentation of DNAJB1-PRKACA-derived HLA class I- and HLA class II-restricted peptides

Three HLA class I-expressing hepatocellular carcinoma (HCC) cell lines (HLE, SMMC-7721, and HepG2; Supplementary Table 3) were transduced with an expression construct that allowed the Doxycycline (Dox)-dependent expression of the DNAJB1-PRKACA fusion protein (Fig. 3a, b, Supplementary Fig. S3a). Subsequent MS-based immunopeptidome analysis revealed up to 3688 different HLA class I ligands (mean 2787; Fig. 3c). Twenty unique HLA class I ligands derived from the DNAJB1-PRKACA fusion protein were identified in the three HCC

cell lines with two ligands spanning the fusion region (Fig. 3d, Supplementary Fig. S3b). The two cellular processed and presented DNAJB1-PRKACA-derived neoepitopes EIFDRYGEEV ($P_{A*68/A*02}$) and IFDRYGEEV ($P_{C*04/C*05}$) identified on SMMC-7721 and HepG2 were predicted to bind to the HLA allotypes HLA-A*68:02 and HLA-C*04:01, respectively, and were validated by comparative spectra analysis using synthetic peptides (Figs. 1c, 3e, Table 1). Of note, the C-terminal AA of $P_{A*68/A*02}$ and $P_{C*04/C*05}$ spanning the fusion region leads to altered HLA presentation of the ligands due to a change in the HLA binding motif anchor position. The corresponding wild-type (WT) peptides would display a glycine at the C-terminal end (EIFDRYGEEG ($P_{A*68/A*02-WT}$) and IFDRYGEEG ($P_{C*04/C*05-WT}$)), which do not allow HLA class I presentation

**Fig. 1 | Prediction of DNAJB1-PRKACA fusion protein-derived HLA class I and HLA class II ligands and characterization of DNAJB1-PRKACA-derived T cell epitopes. a** Overview of the DNAJB1-PRKACA fusion transcript with exon 1 from DNAJB1 and exon 2–10 from PRKACA. **b** HLA class II ligands from the DNAJB1-PRKACA protein fusion region indicating in gray and black in silico predicted and MS-identified ligands, respectively. Continuous lines illustrate the 9 amino acid binding cores, dashed lines the up to 15mer HLA ligand extensions. Allele numbers depict the number of HLA alleles, which are predicted to bind the respective core sequence. **c** HLA class I ligands from the DNAJB1-PRKACA protein fusion region indicating in gray in silico predicted, in black-bordered HLA refolded, and in black MS-identified ligands. **d** HLA allotype population coverage with predicted HLA class I epitopes within the long $P_{II-1}$ of the DNAJB1-PRKACA fusion compared to the European population. The frequencies of individuals within the European population carrying up to six HLA class I allotypes (x-axis) are indicated as gray bars on the left y-axis. The cumulative percentage of population coverage is depicted as black

dots on the right y-axis. **e** Representative example of flow cytometry-based functional characterization ($n = 3$) with indicated cytokines and surface markers of $P_{II-1}$-specific CD4+ T cells derived from a healthy volunteer (HV) 8 after de novo priming with $P_{II-1}$-loaded mature moDCs (upper panel). The negative control presents $P_{II-1}$-primed CD4+ T cells stimulated with a negative peptide (lower panel). **f, g** Absolute frequencies of peptide-specific CD8+ T cells of CD8+ T cells primed with the $P_{A*24}$ or the $P_{A*68/A*02}$, and CD8+ T cells primed with an HLA-matched negative peptide, each dot represents the absolute frequency in one primed well. **f** $P_{A*24}$-specific CD8+ T cells of HV1, HV2, HV6, HV7, and HV9 ($n = 5$). **g** $P_{A*68/A*02}$-specific CD8+ T cells of HV10 ($n = 1$). For boxplots, all data points are shown, the band indicates the median, and the box indicates the first and third quartiles. **h, i** Representative example of IFN-γ and TNF production, as well as CD107a expression of peptide-specific CD8+ T cells after aAPC-priming, stimulated with an HLA-matched negative peptide (lower panel) compared to **h,** $P_{A*24}$ for HV2 ($n = 8$) or **i,** $P_{A*68/A*02}$ for HV10 ($n = 1$) (upper panel). Source data are provided as a Source Data file.

## Table 1 | Predicted DNAJB1-PRKACA fusion protein-derived HLA class I ligands

| Sequence[a] | Peptide ID[b] | HLA restriction[c] | SYFPEITHI score [% of max. score][d] | NetMHCpan [rank][e] | Refolding[f] |
|---|---|---|---|---|---|
| DRYGEEVKEF | | A*24:02 | 36.67 | 0.793 | n/a |
| | | C*06:02 | 58.33 | 1.391 | n/a |
| | | C*07:02 | 66.67 | 1.677 | n/a |
| EEVKEFLA | | B*18:01 | 50.00 | 1.015 | n/a |
| EEVKEFLAK | | B*18:01 | 33.33 | 0.525 | n/a |
| | | B*44:02 | 20.45 | 1.790 | n/a |
| EEVKEFLAKA | $P_{B*44}$ | B*18:01 | n/a | 1.615 | n/a |
| | | B*44:02[g] | 38.89 | 2.816 | Negative |
| **EIFDRYGEEV** | $P_{A*68/A*02}$ | A*02:01[g] | 55.88 | 4.019 | Negative |
| | | A*68:02 | n/a | 0.106 | Positive |
| EIFDRYGEEVK | | A*03:01 | 68.75 | 9.908 | Negative |
| | | A*11:01 | 60.61 | 7.998 | Negative |
| EVKEFLAKAK | | A*03:01 | 67.74 | 3.918 | Negative |
| | | A*11:01 | 63.64 | 3.389 | Negative |
| FDRYGEEVKEF | | A*24:02 | 36.67 | 0.793 | n/a |
| **IFDRYGEEV** | $P_{C*04/C*05}$ | C*04:01 | 80.00 | 0.018 | n/a |
| | | C*05:01 | 66.67 | 0.413 | n/a |
| IFDRYGEEVKEF | | A*24:02 | n/a | 0.469 | n/a |
| | | C*04:01 | n/a | 0.506 | n/a |
| RYGEEVKEF | $P_{A*24}$ | A*24:02 | 74.19 | 0.018 | Positive |
| | | C*04:01 | 56.67 | 0.468 | n/a |
| | | C*06:02 | 53.57 | 1.076 | n/a |
| | | C*07:02 | 80.00 | 0.235 | n/a |
| RYGEEVKEFL | | A*24:02 | 76.67 | 0.462 | n/a |
| | | C*04:01 | 54.84 | 1.011 | n/a |
| YGEEVKEFL | | C*04:01 | 56.67 | 0.610 | n/a |
| | | C*05:01 | 56.67 | 0.451 | n/a |

In silico predicted and mass spectrometric identified (bold) HLA class I ligands derived from the DNAJB1-PRKACA fusion protein. Predictions were performed with SYFPEITHI 1.0 and NetMHCpan 4.1 for the 20 most frequent HLA class I allotypes in the European population (tools.iedb.org).

*ID* identity, *max.* maximum, *n/a* not applicable.

[a]HLA class I ligand amino acid sequence.
[b]Abbreviated peptide name.
[c]Best predicted HLA allele.
[d]Highest binding prediction score with SYFPEITHI, stated in percent of the highest possible score of the respective allele.
[e]Best NetMHCpan binding prediction score.
[f]Indication of whether an HLA refolding experiment was conducted and if it gave a positive or a negative result.
[g]HLA class I ligand with the best available prediction score for the HLA allotype of the FL-HCC patient vaccinated with a personalized DNAJB1-PRKACA-derived peptide vaccine.

of the two WT-peptides on the respective as well as on any other HLA allotype according to netMHC-4.0 and SYFPEITHI predictions. To investigate the processing and presentation of DNAJB1-PRKACA-derived HLA class II peptides, mature moDCs of three HVs were incubated with lysate of the HLE cell line after activation of DNAJB1-PRKACA

fusion gene expression (Supplementary Fig. S3c). MS-based immunopeptidome analysis of these moDCs revealed up to 8293 different HLA class II peptides (mean 5956; Fig. 3f). Thirteen unique peptides derived from the DNAJB1-PRKACA fusion protein were identified with one peptide EVKEFLAKAKEDFLKK ($P_{II-2}$) spanning the fusion region

**Table 2 | HLA class II-presented length variants of $P_{II-1}$ as identified by MS**

| Sequence[a] | Peptide ID[b] | Length [AA][c] | HLA restriction[d] | Binding core[e] | NetMHCIIpan [rank][f] |
|---|---|---|---|---|---|
| FDRYGEEVKEFLAKAKED | | 18 | DPA1*01:03-DPB1*05:01 | EEVKEFLAK | 4.49 |
| REIFDRYGEEVKEFLAKAKED | | 21 | DPA1*01:03-DPB1*05:01 | RYGEEVKEF | 14.23 |
| KREIFDRYGEEVKEFLAKAKED | $P_{II-1}$ | 22 | DPA1*01:03-DPB1*05:01 | RYGEEVKEF | 11.42 |
| IFDRYGEEVKEFLAKAKED | | 19 | DPA1*01:03-DPB1*05:01 | RYGEEVKEF | 5.60 |
| FDRYGEEVKEFLAK | | 14 | DPA1*01:03-DPB1*05:01 | RYGEEVKEF | 1.71 |
| EIFDRYGEEVKEFLAK | | 16 | DPA1*01:03-DPB1*05:01 | RYGEEVKEF | 0.75 |
| EIFDRYGEEVKEFLAKAKED | | 20 | DPA1*01:03-DPB1*05:01 | RYGEEVKEF | 7.84 |
| IFDRYGEEVKEFLAK | | 15 | DPA1*01:03-DPB1*05:01 | RYGEEVKEF | 0.74 |
| REIFDRYGEEVKEFLAK | | 17 | DPA1*01:03-DPB1*05:01 | RYGEEVKEF | 1.12 |
| DRYGEEVKEFLAK | | 13 | DPA1*01:03-DPB1*05:01 | RYGEEVKEF | 8.48 |
| EIFDRYGEEVKEFLAKA | | 17 | DPA1*01:03-DPB1*05:01 | RYGEEVKEF | 1.33 |
| EIFDRYGEEVKEFLAKAK | | 18 | DPA1*01:03-DPB1*05:01 | RYGEEVKEF | 2.86 |
| EIFDRYGEEVKEFLA | | 15 | DPA1*01:03-DPB1*05:01 | RYGEEVKEF | 1.77 |

$P_{II-1}$-derived HLA class II-presented length variants of experimentally eluted $P_{II-1}$-loaded mature monocyte-derived dendritic cells (moDCs) identified by mass spectrometry-based immunopeptidomics.
*AA* amino acid.
[a]HLA class II peptide amino acid sequence.
[b]Abbreviated peptide name.
[c]Peptide amino acid length.
[d]Best predicted HLA class II allele.
[e]Binding core of the best-predicted HLA class II allele.
[f]Best NetMHCIIpan binding prediction score.

(Figs. 1b and 3g). The DNAJB1-PRKACA-derived neoepitope $P_{II-2}$ was predicted to bind to the HLA allele DRB1*13:02 of the respective HV3 (Fig.1b, Supplementary Table 1). The experimental fragment ion spectrum of the $P_{II-2}$ peptide was validated with an isotope-labeled synthetic peptide (Fig. 3h). No HLA class I or HLA class II ligands derived either from the two proteins DNAJB1 and PRKACA or the fusion protein were identified in the respective negative controls.

**Personalized DNAJB1-PRKACA-derived peptide vaccine induces long-lasting DNAJB1-PRKACA-specific immune response and shows favorable clinical outcome in a FL-HCC patient**

A personalized DNAJB1-PRKACA-derived peptide vaccine was designed for a young patient with histologically confirmed FL-HCC (FL-HCC01), who suffered from multiple tumor relapses after receiving an early liver transplant (LTx), due to unresectable FL-HCC not responsive to chemotherapy (Fig. 4a and Supplementary Fig. S4a). The mTOR inhibitor everolimus was applied for post-transplant immunosuppression. Poly (ADP-ribose) polymerase (PARP) inhibition was initiated based on detectable alterations in the DNA damage response (DDR) pathway (ATM and CHEK2, germline variant, BRCA2 and BAP1 somatic deletion)[31], but without achieving a durable remission. Recurrent tumor manifestations were resected or treated with radiotherapy. Based on the prevalence of the DNAJB1-PRKACA fusion gene (confirmed by Sanger sequencing) a vaccine consisting of three short allotype-matching HLA class I ligands (EIFDRYGEEV ($P_{A*68/A*02}$), EEVKEFLAKA ($P_{B*44}$), and IFDRYGEEV ($P_{C*04/C*05}$)), together with the long peptide KREIFDRYGEEVKEFLAKAKED ($P_{II-1}$) predicted to bind to the HLA-DP allotype DPA1*01:03-DPB1*06:01 of FL-HCC01 was composed (Supplementary Fig. S4b and Supplementary Tables 1 and 4). The vaccine was applied twice within a 6-week interval and adjuvanted with the toll-like receptor (TLR) 1/2 agonist XS15 (Pam₃Cys-GDPKHPKSF) emulsified in Montanide™ ISA51 VG to endorse activation and maturation of antigen-presenting cells and prevent vaccine peptides from immediate degradation, thereby enabling induction of an effective and potent T cell response[32–35]. The patient mounted a profound T cell response targeting the $P_{II-1}$ peptide documented 6 weeks after the second vaccination, as analyzed by IFN-γ enzyme-linked immunospot (ELISPOT) assay after in vitro stimulation with the vaccine cocktail peptides (1 mean spot count prior to vaccination versus 872 spot

counts post second vaccination; Fig. 4b, c and Supplementary Fig. S4c). In addition, a weak induction of an IFN-γ T cell response was detected after $P_{B*44}$ stimulation in the ELISPOT assay (0 mean spot count prior to vaccination versus 56 spot counts post second vaccination). Flow cytometry-based characterization of the $P_{B*44}$- and $P_{II-1}$-directed T cell responses revealed T helper 1 (Th1) phenotype CD4⁺ T cells with specific expression of IFN-γ and TNF, whereas no CD8⁺ T cell-driven response against the $P_{II-1}$ or the $P_{B*44}$ was observed, suggesting the $P_{B*44}$-induced IFN-γ T cell response in the ELISPOT assay as cross-reactivity to the shared binding core of the long $P_{II-1}$ (Fig. 4d, e). No T cell responses targeting the $P_{A*68/A*02}$ and $P_{C*04/C*05}$ were observed. Longitudinal IFN-γ ELISPOT assays showed the persistence of $P_{II-1}$-specific T cells over time with a constant intensity of response (770 mean spot count) 18 months after the second vaccination (Fig. 4c and Supplementary Fig. S4c). In contrast to the disease prior to vaccination, where the patient regularly suffered from relapses (time to next relapse 2.5–5.0 months (median 3.0 months)), up to now no disease relapse was observed (21 months after the second vaccination) pointing to the clinical efficacy of vaccine-induced DNAJB1-PRKACA-specific T cell responses.

In order to identify the vaccine-induced T cell clones expanding in FL-HCC01 post-vaccination, combined single-cell RNA and single-cell TCR sequencing utilizing 10x Genomics single-cell immune profiling was performed. Unsupervised clustering of single-cell RNA sequencing data from vaccine-induced $P_{II-1}$-reactive CD4⁺ T cells after in vitro expansion, followed by flow-cytometry-based bulk sort of CD4⁺ T cells, showed three defined T cell clusters: (I) cytokine and chemokine-expressing activated T cells defined by high expression of IFNG, TNF, GZMB (encoding granzyme B), CCL3, and CCL4 (Fig. 5a–c), (II) T cells exhibiting an exhausted or late effector profile with an expression of PDCD1, LAG3, HAVCR2, and CTLA4 (Fig. 5a, b and Supplementary Fig. S5a), and (III) naïve resting T cells defined by expression of SELL (encoding CD62L), CCR7, and TCF7 (Fig. 5a, b and Supplementary Fig. S5b). In agreement, functional enrichment for the hallmarks of cancer gene sets "TNF signaling via NF-κB" and "inflammatory response" showed an increased gene expression in the cytokine and chemokine expressing activated T cell cluster (Fig. 5d), indicating that this cluster comprised the CD4⁺ T cells reactive to the $P_{II-1}$. As expected, VDJ sequencing revealed a high TCR clonality in the activated T cell

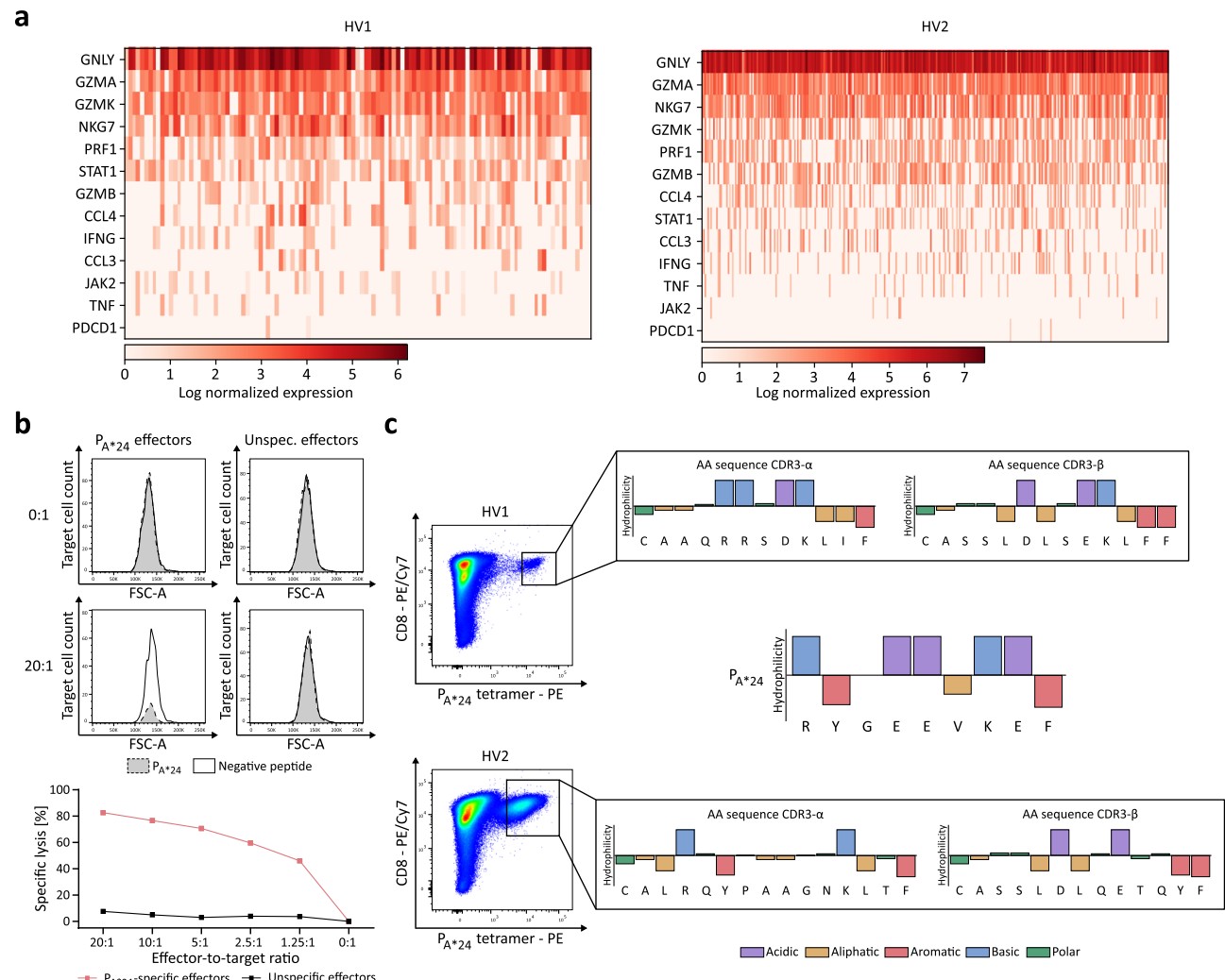

**Fig. 2 | In-depth characterization of DNAJB1-PRKACA-specific CD8⁺ T cells and single-cell TCR sequencing. a** Heat map of single-cell RNA sequencing analysis of flow cytometry-based bulk sorted P_{A*24}-specific CD8⁺ T cells of a healthy volunteer (HV) 1 and HV2 after aAPC-based priming with HLA-A*24-P_{A*24}-monomer, showing log normalized gene expression for selected activated T cell markers. **b** Specific cell lysis by P_{A*24}-specific CD8⁺ T cells from HV2, of P_{A*24}-loaded autologous CD8⁻ target cells (gray fill, dashed line (upper panel); red line (lower panel)) at various effector-to-target cell ratios compared to negative peptide-loaded autologous CD8⁻ target cells (white fill, solid line (upper panel)). P_{A*24}-unspecific CD8⁺ T cells showed no lysis of the target cells (black line (lower panel). Results are shown for three independent technical replicates. **c** Flow cytometry-based bulk sort of P_{A*24}-specific CD8⁺ T cells of two HVs (HV1, HV2) after aAPC-based priming with HLA-A*24-P_{A*24}-monomer (left panel) for single-cell T cell receptor (TCR) sequencing. The right panel depicts physiochemical properties and amino acid sequences of the CDR3-α/-β region of the most frequent TCR clone from each donor in comparison to their target peptide P_{A*24}. On the y-axes, the hydrophilicity according to the Hopp-Woods scale[82] is indicated and amino acids (AA) are grouped by their physiochemical properties with color code. Source data are provided as a Source Data file.

cluster with 74.2% of cells assigned to large clones (clonality ≥ 4), compared to the naïve resting (1.3%) or exhausted T cell clusters (31.4%) (Fig. 5e and Supplementary Fig. S5c). In total, 10 defined TCR clones were identified, of which eight were predominantly assigned to the activated T cell cluster (Fig. 5f, Supplementary Table 5). The high similarity of physiochemical properties and hydrophilicity of the AA sequences was observed for the CDR3-α/-β sequences of the ten expanded TCR clones, especially for positions four and five of the CDR3-α sequences with opposing characteristics regarding chemical groups in comparison to the target peptide core-binding motif (Fig. 5g, Supplementary Fig. S5d and Supplementary Table 5). By clustering the CDR3-α/-β sequences distinct motif plots were generated for the 10 expanded TCR clones in comparison to the naïve unexpanded TCRs, which showed a specific preference for basic AAs on position five of the CDR3-α sequence which could not be observed in the unexpanded clones (no significant Pearson correlation coefficient to the negative cluster). Specific differences (no significant motif correlation) between the expanded clones and the negative dataset were also observed for

position six of the CDR3-α cluster and position six, seven, and eight of the CDR3-β motif (Fig. 5h).

## Discussion

T cell recognition of HLA-presented antigens plays a central role in the immune surveillance of malignant disease[36,37]. Numerous immunotherapeutic approaches aim to utilize respective tumor antigens to therapeutically induce an anti-tumor T cell response[7,8,17,18]. This study reports on the identification and characterization of immunogenic neoantigens derived from the DNAJB1-PRKACA fusion transcript, which is the oncogenic driver in all patients suffering from FL-HCC[24,25]. Neoepitopes derived from oncogenic gene fusions have been suggested as a superior category of tumor antigens[23]. This has been attributed to (I) the clonal expression of oncogenic driver gene fusions[23,38], (II) the higher degree of sequence alteration compared to somatic point mutations, resulting in increased immunogenicity[39], and (III) limited down-regulated-based immune escape[40]. In contrast to other fusion transcripts[41–43], the DNAJB1-PRKACA fusion generates a

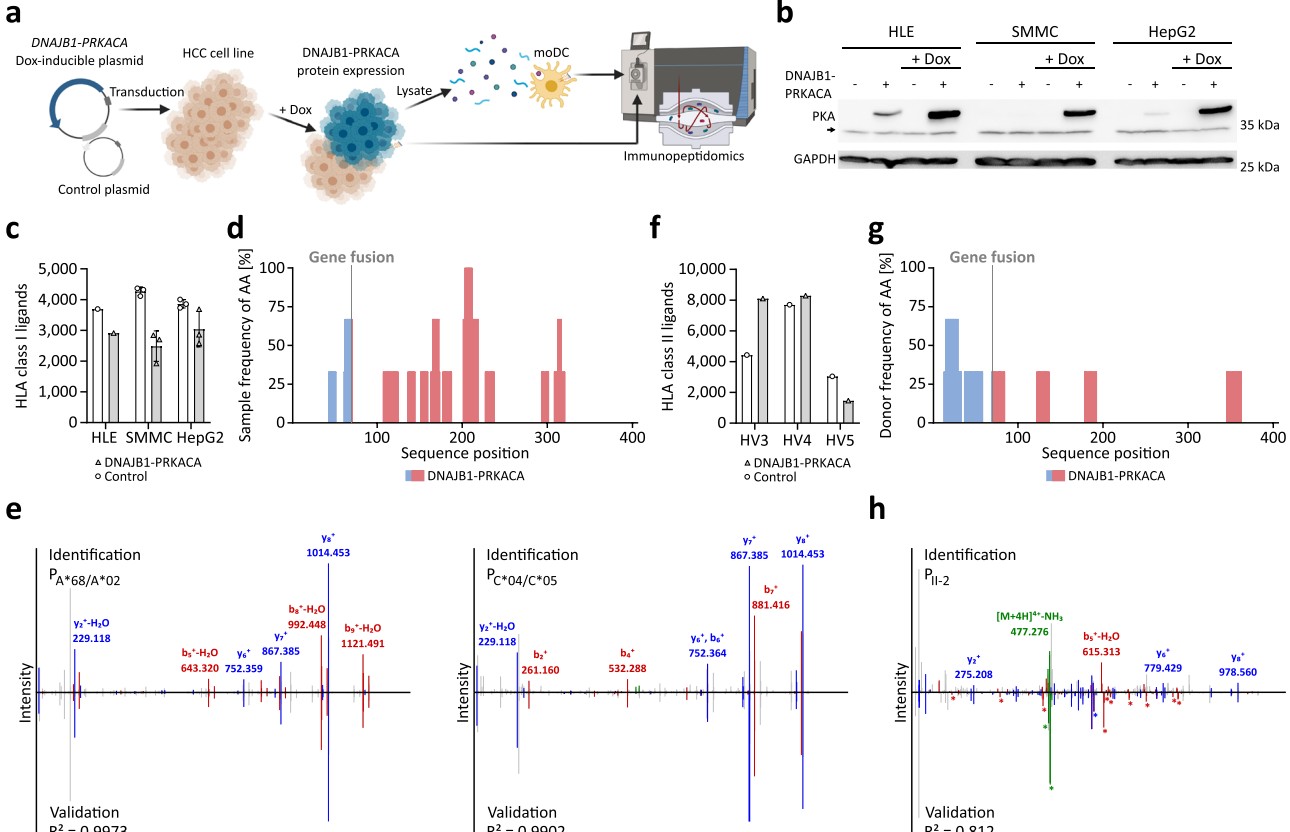

**Fig. 3 | Mass spectrometry-based identification of naturally presented DNAJB1-PRKACA-derived HLA class I and HLA class II ligands. a** Schematic overview of Doxycycline (Dox)-inducible DNAJB1-PRKACA fusion gene expression. Hepatocellular carcinoma (HCC) cell lines transduced with DNAJB1-PRKACA Dox-inducible or control plasmid were treated with Dox followed by mass spectrometry (MS)-based immunopeptidome analysis of the HCC cell line, or of mature monocyte-derived dendritic cells (moDC) of a healthy volunteer (HV) incubated with HCC cell line lysate (created with BioRender.com). **b** Dox-induced DNAJB1-PRKACA expression by immunoblotting of HCC cell lines (HLE, SMMC-7721, and HepG2) carrying the DNAJB1-PRKACA Dox-inducible (+) or the control plasmid (−) with and without Dox treatment using an anti-Protein Kinase A (PKA) antibody ($n = 3$). The black arrow indicates the endogenous PKA band. GAPDH served as a loading control. **c, f** MS-identified peptides of **c** HCC cell lines (HLE $n = 1$, SMMC-7721 $n = 3$, and HepG2 $n = 3$) carrying the DNAJB1-PRKACA Dox-inducible or the control plasmid after Dox-treatment (data are presented as mean values ± SD) and **f** of mature moDCs of HVs incubated with HCC cell line lysate with or without expression of DNAJB1-PRKACA protein, respectively. **d, g** Distribution of MS-identified HLA ligands over the DNAJB1-PRKACA fusion protein sequence plotted with the frequency of amino acids (AA) per sample for, **d** HLA class I ligands of the HCC cell lines ($n = 3$) expressing DNAJB1-PRKACA and **g** HLA class II peptides of mature moDCs of HV3, HV4, and HV5 ($n = 3$) incubated with HCC cell line lysate expressing DNAJB1-PRKACA. **e, h** Fragment spectra ($m/z$ on the $x$-axis) of the experimentally eluted peptides (**e**) of experimentally eluted HLA class I-presented ligands EIFDRYGEEV ($P_{A*68/A*02}$, left) and IFDRYGEEV ($P_{C*04/C*05}$, right) extracted from the DNAJB1-PRKACA expressing cell lines SMMC-7721 and HepG2, respectively (identification) by comparison to the respective synthetic peptide (validation, mirrored on the $x$-axis) with the calculated spectral correlation coefficient ($R^2$). **h** Validation of the experimentally eluted HLA class II-presented peptide EVKEFLAKAKEDFLKK ($P_{II-2}$) extracted from mature moDCs of a HV (identification) by comparison to the respective synthetic peptide isotope labeled on AA position two (validation, mirrored on the $x$-axis) with the calculated spectral correlation coefficient ($R^2$). Identified b- and y-ions are marked in red and blue, respectively. Isotope-labeled ions are marked with asterisks. Source data are provided as a Source Data file.

defined and unique protein sequence that allows an off-the-shelf application of DNAJB1-PRKACA-derived neoepitopes in cancer immunotherapy. We validated the DNAJB1-PRKACA fusion protein as a source of immunogenic HLA class I and HLA class II-binding antigens inducing both CD8[+] and CD4[+] T cell responses, which is required for effective anti-cancer immunity[44,45]. Furthermore, we proved the cellular processing and HLA-restricted presentation of DNAJB1-PRKACA neoepitopes, which is an indispensable prerequisite for therapeutically used tumor antigens in particular regarding the distorted correlation between gene expression and HLA-restricted antigen presentation, with only a small fraction of alterations on DNA level resulting in an HLA-presented neoepitope on the tumor cell surface[13,22,46,47]. However, for the immunogenic neoepitope $P_{A*24}$ natural processing and presentation could not be validated in the DNAJB1-PRKACA transduced HCC cell lines, within the sensitivity limitation of the current state-of-the-art mass spectrometry-based immunopeptidomics. The sensitivity of shotgun mass spectrometric discovery approaches is, even in the

context of immense technical improvements in the last decades, still limited[48]. Therefore, we cannot exclude the low-level presentation of the $P_{A*24}$.

We further report on the clinical application of a DNAJB1-PRKACA neoepitope-based personalized peptide vaccine adjuvanted with the TLR1/2 agonist XS15[32] and Montanide™ ISA51 VG in a single FL-HCC patient. We observed profound and long-lasting DNAJB1-PRKACA-specific T cell responses showing a clonal expansion of activated CD4[+] T cells, despite ongoing mTOR inhibition-based immunosuppression with contradictory reported effects on T cells[49,50]. Follow-up data until month 18 after vaccination showed the persistence of profound DNAJB1-PRKACA-specific T cell responses. This is in line with the induction of long-term virus-specific T cell responses observed upon XS15-adjuvanted multi-peptide vaccination with our CoVac-1 COVID19 vaccine candidate[35]. Induction of long-lasting T cell responses was mirrored by so far relapse-free survival of the patient, indicating, in line with other tumor vaccines[17,18], a potential of DNAJB1-PRKACA-based

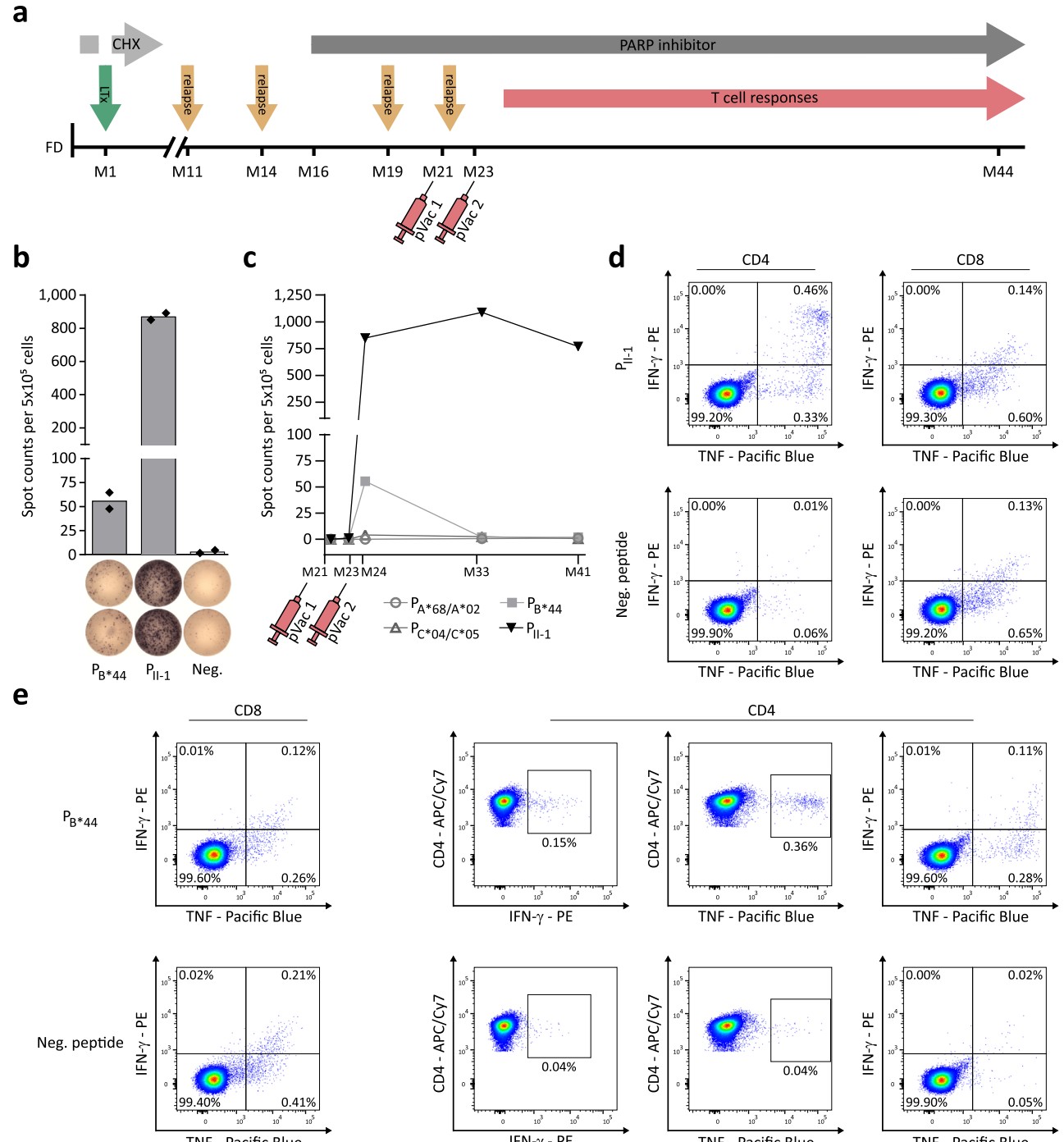

**Fig. 4 | FL-HCC patient vaccinated with a personalized DNAJB1-PRKACA-derived peptide vaccine. a** Schematic therapy course of a fibrolamellar hepatocellular carcinoma (FL-HCC) patient treated with a DNAJB1-PRKACA-derived peptide vaccine cocktail. After first diagnosis (FD) the patient was treated with four cycles of chemotherapy (CHX) analogous to the PHITT study (PHITT Group F) interrupted by an early liver transplant (LTx) one month (M1) after FD, as the tumor was assessed not resectable. Everolimus was used for post-transplantation immunosuppression. The patient experienced four relapses after LTx at months 11, 15, 19, and 21 post-FD. Tumor manifestations of the first, second, and fourth relapse were surgically resected, and for the third relapse, radiotherapy was applied. Starting at month 16 post-FD the patient was treated with Olaparib (poly (ADP-ribose) polymerase (PARP) inhibitor). At months 21 and 23, the patient received two vaccinations of a personalized DNAJB1-PRKACA-derived peptide vaccine comprising the peptides $P_{A*68/A*02}$, $P_{B*44}$, $P_{C*04/C*05}$, and $P_{II-1}$. Induction of vaccine peptide-specific T cell responses was observed six weeks after the second vaccination. **b** Vaccine peptide-specific T cell responses 6 weeks after the second vaccination were assessed by IFN-γ ELISPOT assay after in vitro stimulation with the vaccine cocktail peptides ($P_{B*44}$, $P_{II-1}$) compared to the negative peptide (neg.). **c** Longitudinal analysis of vaccine-induced T cell responses up to 18 months post-vaccination using IFN-γ ELISPOT assay after in vitro stimulation with the vaccine cocktail peptides ($P_{A*68/A*02}$, $P_{B*44}$, $P_{C*04/C*05}$, and $P_{II-1}$) in technical duplicates. **d**, **e** Flow cytometry-based characterization for indicated cytokines of **d**, CD4$^+$ and CD8$^+$ T cells stimulated with the $P_{II-1}$ peptide and **e**, CD4$^+$ and CD8$^+$ T cells stimulated with the $P_{B*44}$ peptide 14 weeks after the second vaccination in comparison to the respective negative (neg.) peptides. Source data are provided as a Source Data file.

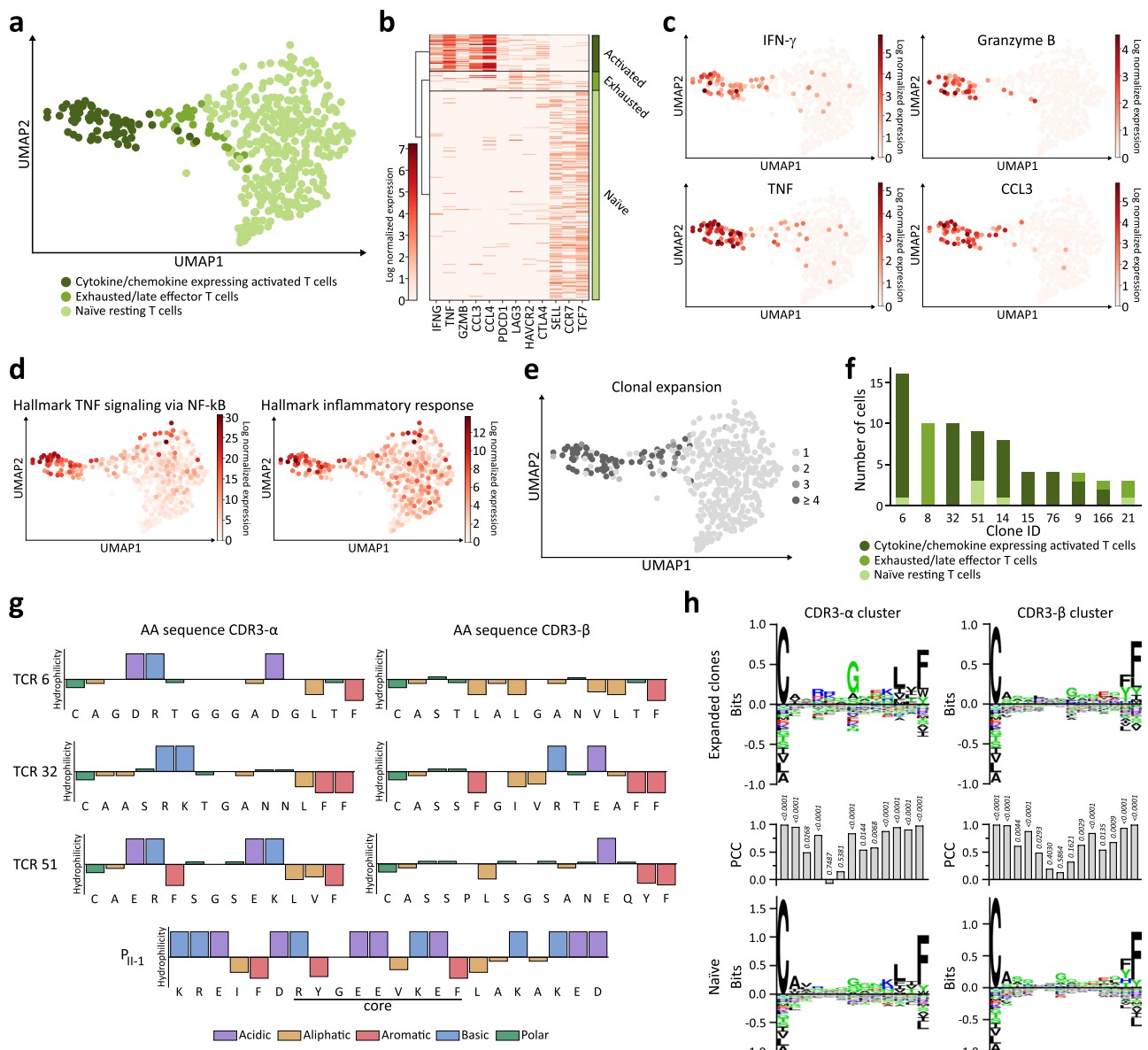

**Fig. 5 | Single-cell RNA sequencing of vaccine-induced $P_{II\text{-}1}$ specific CD4$^+$ T cells.**
**a–f** Single-cell RNA-sequencing analysis of CD4$^+$ T cells sorted from $P_{II\text{-}1}$-stimulated PBMCs of the FL-HCC patient 31 weeks after the second vaccination with a personalized DNAJB1-PRKACA-derived peptide vaccine. **a** Uniform Manifold Approximation and Projection (UMAP) plot showing distinct T cell clusters. **b** Heat map of cluster defining log normalized gene expression for activated, exhausted/late effector, and naïve resting T cells. **c** UMAP plots depicting log normalized IFNG, GZMB (encoding granzyme B), TNF, and CCL3 gene expression defining the activated T cell cluster. **d** Functional enrichment of the log normalized gene expression for the hallmark cancer gene sets "TNF signaling via NF-κB" and "inflammatory response". **e** UMAP plot showing T cell receptor (TCR) clonality of sequenced CD4$^+$ T cells. The color code indicates the number of cells belonging to an expanded clonotype. **f** Distribution of the 10 largest TCR clonotypes across the cell type clusters identified in $P_{II\text{-}1}$-specific CD4$^+$ T cells. **g** Physiochemical properties and amino acid (AA) sequences of the CDR3-α/-β region of the most frequent TCR clones in comparison to their target peptide $P_{II\text{-}1}$. The *y*-axes indicate the hydrophilicity according to the Hopp-Woods scale[82]. AAs are grouped by their physiochemical properties with color code. **h** Clustering of variable sequences of the CDR3-α and CDR3-β region of the ten largest TCR clonotypes across the cell type clusters identified in $P_{II\text{-}1}$-specific CD4$^+$ T cells (upper panel), compared to all identified naïve single cell clones (lower panel). Position-wise Pearson correlation coefficient (PCC) calculation between positive and negative dataset (middle panel), with the significance of correlation calculated using a two-sided Pearson correlation test. The clustering was conducted using GibbsCluster 2.0[80]. Source data are provided as a Source Data file.

vaccines to combat residual tumor cells. Peptide vaccination of the FL-HCC patient was applied under continued PARP inhibitor treatment, which started 5 months prior to vaccination and did not prevent the occurrence of relapse applied as a single substance. The combination of PARP inhibition and DNAJB1-PRKACA neoepitope-based peptide vaccine could have positively affected the vaccination response and the prolonged relapse-free survival of the FL-HCC patient based on the beneficial impact of DNA-damaging agents on tumor immunogenicity, which was previously reported for the combination of PARP inhibition

and ICIs[51,52]. We could not detect any vaccine-induced CD8$^+$ T cell responses against the DNAJB1-PRKACA fusion protein. Thus, tumor immune surveillance and relapse-free survival observed in the patient after vaccination might be mediated by DNAJB1-PRKACA-specific CD4$^+$ T cells alone[53], or accompanied by undetected CD8$^+$ T cells recognizing other tumor antigens that were induced by the CD4$^+$ T cells via epitope spreading[54]. Moreover, the in vitro expansion of patient-derived T cells prior to characterization by flow cytometry and single-cell RNA sequencing might have impacted the phenotypes of these cells.

This low side-effect peptide vaccine represents so far the only DNAJB1-PRKACA-targeted therapy[30] and might in the future be applied within combinatorial treatment approaches comprising newly developed small molecules targeting the kinase activity of PRKACA[30] and/or ICIs. Combination with the latter is supported by (I) the high expression of PD-L1 in FL-HCC[55], (II) the detection and response correlation of fusion protein-specific T cells in patients receiving ICIs[23], and (III) the first promising clinical results of neoepitope-based vaccines in combination with ICIs[17,18]. Beyond the design of vaccines, DNAJB1-PRKACA-derived neoepitopes could serve as targets for the development of therapeutic approaches using adoptive T cell transfer and TCR engineering. For this purpose, we identified multiple TCRs from in vitro and in vivo induced DNAJB1-PRKACA-specific T cells displaying a unique basic AA motif of the CDR3-α sequence with opposing characteristics regarding chemical groups in comparison to the target peptide core-binding motif[56].

FL-HCC is a rare tumor disease; however recent reports state that the number of FL-HCC cases might be significantly underdiagnosed[57,58], suggesting a growing number of cases in the future. Furthermore, recent advances in genome sequence analysis[59,60] enable the identification of further cancer entities that express the DNAJB1-PRKACA fusion transcript[30]. This study identifies the DNAJB1-PRKACA fusion transcript as a prime source for broadly applicable neoepitopes and provides evidence for their immunotherapeutic efficacy in a single FL-HCC patient. Open questions remain if future off-the-shelf T cell-based immunotherapies targeting the DNAJB1-PRKACA fusion will be able to overcome the immunosuppressive microenvironment and other escape mechanisms of tumors[61] to natural immune surveillance and which of the multiple predicted HLA allotypes give rise to DNAJB1-PRKACA neoepitopes in vivo and thus which patients will profit from, e.g. a neoepitope-based vaccine. These issues will be addressed in an upcoming clinical trial evaluating the here-defined DNAJB1-PRKACA neoepitopes adjuvanted with the TLR1/2 agonist XS15 emulsified in Montanide™ ISA51 VG[32,35] in combination with the PD-L1 antibody atezolizumab in a Phase I vaccine study, recruiting patients with various malignant disease expressing the DNAJB1-PRKACA fusion transcript.

## Methods

The study was performed according to the guidelines of the ethics committee at the medical faculty of the Eberhard-Karls-University and at the University Hospital Tübingen (713/2018BO2, 406/2019BO2).

### Patients and blood samples

Peripheral blood mononuclear cells (PBMCs) from the FL-HCC patient ($n = 1$) as well as PBMCs from healthy volunteers (HVs, $n = 11$) were isolated by density gradient centrifugation and stored at −80 °C until further use for subsequent T cell-based assays. Informed consent was obtained in accordance with the Declaration of Helsinki protocol, and donors were not financially compensated. The study was performed according to the guidelines of the local ethics committees (713/2018BO2, 406/2019BO2). HLA typing was carried out by the Department of Hematology and Oncology, Tübingen, Germany (Supplementary Table 1).

### Personalized peptide vaccine

The personalized vaccine developed and produced by the Good Manufacturing Practices (GMP) Peptide Laboratory of the Department of Immunology, University Tübingen, Germany, is a peptide-based vaccine containing four DNAJB1-PRKACA-derived peptides (Supplementary Table 4) and the adjuvant lipopeptide synthetic TLR1/2 ligand XS15[8] (manufactured by Bachem AG, Bubendorf, Switzerland) emulsified in Montanide™ ISA51 VG[9] (manufactured by Seppic, Paris, France). Vaccine peptides (250 μg/peptide) and XS15 (50 μg) are prepared as a water–oil emulsion 1:1 with

Montanide™ ISA51 VG to yield an injectable volume of 500 μl. Following disclosure and written consent, the patient received a subcutaneous injection of the personalized vaccine in the lower abdomen. The treatment of the patient was performed within a compassionate use program (expanded access) for personalized peptide vaccination under the project: clinicaltrials.gov NCT05014607. The local Ethics Committee (406/2019BO2) and the regulatory authority (Regierungspräsidium Tübingen) approved the project which was conducted under the German Drug Law §13 paragraph 2b. The patient gave written informed consent for vaccine treatment, sequential blood analysis for immunomonitoring, as well as publishing of the related data.

### Detection of DNAJB1-PRKACA transcript and sequencing

RNA was extracted from macrodissected 5 μm paraffin sections using the Maxwell® RSC RNA FFPE Kit and the Maxwell® RSC Instrument (Promega, Madison, WI, USA) according to the manufacturer's instructions. Reverse transcription of RNA and polymerase chain reaction (PCR) of the DNAJB1-PRKACA breakpoint region (forward primer 5′-GTTCAAGGAGATCGCTGAGG-3′, reverse primer 5′- TTCCCGGTCTCCTTGTGTTT-3′) was performed using the QIAGEN OneStep RT-PCR Kit according to the manufacturer's instructions (Qiagen, Hilden, Germany). To visualize the detection of the DNAJB1-PRKACA fusion, the PCR product was run on an agarose gel. For sequencing, the PCR product was purified (AMPure, Beckman Coulter, Brea, CA, USA) and aliquots were used for the sequencing reaction with 1 μM of the forward or reverse primer and 2 μl of GenomeLab DTCS-Quick Start Master Mix (Beckman Coulter, Brea, CA, USA) in a final volume of 10 μl according to the manufacturer's protocol. Sequencing reactions were purified (CleanSEQ, Beckman Coulter, Brea, CA, USA) and analyzed in a GenomeLab GeXP Genetic Analysis System, and evaluated by the GenomeLab GeXP software (Beckman Coulter, Brea, CA, USA).

### Histology and immunohistochemistry

Tissue specimens were obtained during the routine diagnostic procedure, fixed in 4% formalin, and embedded in paraffin (FFPE). 1.5–3.0 μm-thick sections were cut by using a microtome and stained with haematoxylin and eosin (HE) or with Masson's trichrome as additional routine staining for liver specimens. Immunohistochemistry was performed on an automated immunostainer (VENTANA Bench-Mark ULTRA, Ventana Medical Systems, Oro Valley, AZ, USA) according to in-house protocols. Slides were stained by using CK7 (1:2000, Clone OV-TL 12/30, Agilent Dako, Santa Clara, CA, USA) and hepatocyte paraffin1 (Hepar1) (1:1000, Clone OCH1E5, Agilent Dako, Santa Clara, CA, USA) as primary antibodies. Slide scans of the hepatectomy specimen were produced by using the Ventana Scanner DP200 (Ventana Medical Systems, Oro Valley, AZ, USA).

### In silico prediction of DNAJB1-PRKACA-derived HLA class I and HLA class II ligands

HLA class I DNAJB1-PRKACA ligand prediction was performed for all possible 8–12 AA long peptide sequences spanning the fusion region using SYFPEITHI 1.0[62] and NetMHCpan 4.1[63] for the 20 most frequent HLA class I allotypes in the European population (tools.iedb.org). HLA class II DNAJB1-PRKACA ligand prediction was performed for all possible 15 AA long peptide sequences spanning the fusion region using NetMHCIIpan 4.0[63] with all listed allele combinations.

### Quantification of HLA surface expression

HLA surface expression of HCC cell lines was analyzed using the QIFIKIT bead-based quantitative flow cytometric assay (Dako, K0078) according to the manufacturer's instructions as described before[64]. In brief, samples were stained with the pan-HLA class I-specific monoclonal antibody (mAb) W6/32 (produced in-house) or IgG isotype

control (BioLegend, 400202). Flow cytometric analysis was performed on a FACSCanto II Analyzer (BD).

## Doxycycline-inducible DNAJB1-PRKACA fusion gene expression in HCC cell lines

The HCC cell lines HLE (obtained from the Japan Collection of Research Bioresources (JCRB) Cell Bank), SMMC-7721 (obtained from Woodland Pharmaceuticals), and HepG2 (obtained from the American Type Culture Collection (ATCC)) were cultivated in Gibco Dulbecco's Modified Eagle Medium supplemented with 10% fetal calf serum (FCS), penicillin, streptomycin (all from Merck) and plasmocin (Invivogen) at 37 °C and 5% $CO_2$ in a humidified atmosphere. The SMMC-7721 cell line is under the list of known misidentified cell lines maintained by the International Cell Line Authentication Committee; however, the cell line was selected due to the specific HLA type and was validated by HLA typing. The DNAJB1-PRKACA coding sequence was synthesized by Thermo Fisher, cloned into the pENTR™ plasmid, and transferred by directional TOPO cloning (pENTR™/D-TOPO™ cloning kit, Invitrogen) into the pInducer20 (Addgene #44012) destination vector. Lentiviral particles were produced in HEK293T cells (obtained from the DSMZ) by calcium-phosphate transfection of the helper plasmids psPAX2, pMD2.G, and either the empty or the DNAJB1-PRKACA coding pInducer20. The transduced HCC cell lines were selected with G418 (Invivogen) for at least 2 weeks. To induce the expression of DNAJB1-PRKACA, the transduced HCC cell lines were treated with 1 µg/ml Doxycycline (Dox) for 24 h (AppliChem). To validate the DNAJB1-PRKACA expression, cells were lysed in lysis buffer (50 mM Tris−HCl pH 7.4, 150 mM NaCl, 1% Triton X-100, 50 mM NaF, 10 mM $Na_4P_2O_7$, 10 mM $Na_4V_2O_7$ and complete protease inhibitor cocktail (Roche)). SDS−PAGE and immunoblotting were performed as described previously[65]. For immunoblotting the primary antibodies anti-PKAα cat (1:1000 dilution, Santa Cruz, clone A-2, Cat# sc-28315, RRID:AB_628136), anti-GAPDH (1:2000 dilution, Cell Signaling, clone D16H11, Cat# 5174, RRID:AB_10622025), and anti-Tubulin (1:2000 dilution, Merck, clone DM1A, Cat# 05-829, RRID:AB_310035) were used. HRP-coupled goat anti-rabbit or goat anti-mouse (both Jackson ImmunoResearch) secondary antibodies were used for visualization. Uncropped and unprocessed scans are supplied in the Source Data file.

## Isolation of HLA ligands

HLA class I and HLA class II molecules were isolated by standard immunoaffinity purification[66] using the pan-HLA class I-specific mAb W6/32, the pan-HLA class II-specific mAb Tü-39, and the HLA-DR-specific mAb L243 (all produced in-house) to extract HLA ligands.

## Analysis of HLA ligands by liquid chromatography−coupled tandem mass spectrometry (LC−MS/MS)

Peptide samples were separated by reversed-phase liquid chromatography (nanoUHPLC, UltiMate 3000 RSLCnano, Thermo Fisher, Waltham, MA, USA) and subsequently analyzed in an online coupled Orbitrap Fusion Lumos mass spectrometer (Thermo Fisher). Samples were analyzed in three technical replicates. Sample volumes of 5 µl with shares of 20% were injected onto a 75 µm × 2 cm trapping column (Thermo Fisher) at 4 µl/min for 5.75 min. Peptide separation was subsequently performed at 50 °C and a flow rate of 300 nL/min on a 50 µm × 25 cm separation column (PepMap C18, Thermo Fisher) applying a gradient ranging from 2.4% to 32.0% of ACN over the course of 90 min. Eluting peptides were ionized by nanospray ionization and analyzed in the mass spectrometer implementing a top speed (3 s) HCD (Higher-energy C-trap dissociation) method generating fragment spectra with a resolution of 30,000, a mass range limited to 235–1151 m/z, and positive charge states 2−5 selected for fragmentation.

## Data processing

Data processing was performed as described previously[67]. The Proteome Discoverer (v1.4, Thermo Fisher) was used to integrate the search results of the SequestHT search engine (University of Washington[68]) against the human proteome (Swiss-Prot database, 20,279 reviewed protein sequences, September 27, 2013) accompanied by the complete sequence of the DNAJB1-PRKACA fusion protein. Precursor mass tolerance was set to 5 ppm and fragment mass tolerance was set to 0.02 Da. Oxidized methionine was allowed as a dynamic modification. The false discovery rate (FDR, estimated by the Percolator algorithm 2.04[69]) was limited to 5% for HLA class I and 1% for HLA class II. HLA class I annotation was performed using SYFPEITHI 1.0[62] and NetMHCpan 4.1[63].

## Spectrum validation

Spectrum validation of the experimentally eluted peptides was performed by computing the similarity of the spectra with corresponding synthetic peptides measured in a complex matrix. The spectral correlation was calculated between the MS/MS spectra of the eluted and the synthetic peptide[70].

## Amplification of peptide-specific T cells and IFN-γ ELISPOT assay

PBMCs were pulsed either with 1 µg/ml or with 5 µg/ml of HLA class I or HLA class II peptide, respectively. Irrelevant peptides with the respective HLA restrictions were used as negative control (YLLPAIVHI for HLA-A*02 (source protein: DDX5_HUMAN), and ETVITVDTKAAGKGK for HLA class II (source protein: FLNA_HUMAN)). Cells were cultured for 12 days adding 20 U/ml IL-2 (Novartis, Basel, Switzerland) on days 2, 5, and 7. Peptide-stimulated PBMCs were analyzed by IFN-γ enzyme-linked immunospot (ELISPOT) assay on day 12[67], with anti-IFN-γ antibody (clone 1-D1K, 2 µg/mL, MabTech), anti-IFN-γ biotinylated detection antibody (clone 7 B6 1, 0.3 µg/mL, MabTech), ExtrAvidin-Alkaline Phosphatase (1:1000 dilution, Sigma-Aldrich) and BCIP/NBT (5 bromo 4-chloro 3 indolyl-phosphate/nitro-blue tetrazolium chloride, Sigma-Aldrich). Spots were counted using an ImmunoSpot S6 analyzer (CTL, Cleveland, OH, USA) and T cell responses were considered positive if >10 spots/500,000 cells were counted, and the mean spot count was at least three-fold higher than the mean spot count of the negative control.

## Refolding

Biotinylated HLA:peptide complexes were manufactured as described previously[71] and tetramerized using PE-conjugated streptavidin (Invitrogen) at a 4:1 molar ratio.

## Induction of peptide-specific CD8+ T cells with aAPCs

Priming of peptide-specific cytotoxic T lymphocytes was conducted using aAPCs as described previously[72]. In detail, 800,000 streptavidin-coated microspheres (Bangs Laboratories, Fishers, IN, USA) were loaded with 200 ng biotinylated HLA:peptide monomer and 600 ng biotinylated anti-human CD28 monoclonal antibody (clone 9.3, in-house production). CD8+ T cells were cultured with 4.8 U/µl IL-2 (R&D Systems, Minneapolis, MN, USA) and 1.25 ng/ml IL-7 (PromoKine, Heidelberg, Germany). Weekly stimulation with aAPCs (200,000 aAPCs per 1 × 10⁶ CD8+ T cells) and 5 ng/ml IL-12 (PromoKine) was performed for four cycles.

## Cytokine, surface marker and tetramer staining

Functionality of peptide-specific CD4+ and CD8+ T cells was analyzed by surface marker and intracellular cytokine staining (ICS) as described previously[73,74]. Cells were pulsed with 10 µg/ml of respective peptide and incubated with 10 µg/ml Brefeldin A (Sigma-Aldrich, Saint Louis, MO, USA) and 10 µg/ml GolgiStop (BD, Franklin Lakes, NJ, USA) for 12–16 h. Staining was performed using Cytofix/Cytoperm (BD), Aqua live/dead (1:400 dilution, Invitrogen), APC/Cy7 anti-human CD4

(1:100 dilution, BioLegend, Cat# 300518, RRID: AB_314086), PE/Cy7 anti-human CD8 (1:400 dilution, Beckman Coulter, Cat# 737661, RRID: AB_1575980), Pacific Blue anti-human TNF (1:120 dilution, BioLegend, Cat# 502920, RRID: AB_528965), FITC anti-human CD107a (1:100 dilution, BioLegend, Cat# 328606, RRID: AB_1186036), APC anti-human IL-2 (1:40 dilution, BioLegend, Cat# 500309, RRID: AB_315096), and PE anti-human IFN-γ mAB (1:200 dilution, BioLegend, Cat# 506507, RRID: AB_315440). PMA and ionomycin (Sigma-Aldrich) served as a positive control. Negative control peptides with matching HLA restrictions were used: YLLPAIVHI for HLA-A*02 (source protein: DDX5_HUMAN), KYPENFFLL for HLA-A*24 (source protein: PP1G_HUMAN), EEFGRAFSF for HLA-B*44 (source protein: HLA-DP_HUMAN), and ETVITVDT-KAAGKGK for HLA class II (source protein: FLNA_HUMAN). Gating strategies applied for the analyses of flow cytometry-acquired data are provided in Supplementary Figs. S6, S7, and S8.

The frequency of peptide-specific CD8+ T cells after aAPC-based priming was determined by Aqua live/dead (1:400 dilution, Invitrogen), PE/Cy7 anti-human CD8 (1:400 dilution, Beckman Coulter, Cat# 737661, RRID: AB_1575980) and HLA:peptide tetramer-PE staining. Cells of the same donor primed with an irrelevant control peptide TYSEKTTLF (source protein: MUC16_HUMAN) and stained with the tetramer containing the test peptide were used as a negative control. The priming was considered successful if the frequency of peptide-specific CD8+ T cells was ≥0.1% of CD8+ T cells within the viable single cell population and at least three-fold higher than the frequency of peptide-specific CD8+ T cells in the negative control. The same evaluation criteria were applied to ICS results. Samples were analyzed on a FACS Canto II cytometer (BD). The gating strategy applied for tetramer staining analysis of flow cytometry-acquired data is provided in Supplementary Fig. S9.

### Cytotoxicity assays

Peptide-specific CD8+ T cells were analyzed for their capacity to induce peptide-specific target cell lysis in the flow cytometry-based VITAL assay[22]. Autologous CD8− target cells were either loaded with the $P_{A*24}$ peptide or the HLA-matched negative peptide KYPENFFLL (source protein: PP1G_HUMAN) and labeled with CFSE or FarRed (Life Technologies, Carlsbad, CA, USA), respectively. The $P_{A*24}$-specific effector cells were added in the indicated effector-to-target ratios. Specific lysis of peptide-loaded CD8− target cells was calculated relative to control targets.

### Induction of peptide-specific CD4+ T cells with peptide-loaded moDCs

For the differentiation of monocyte-derived dendritic cells (moDCs) CD14+ cells were isolated from PBMC using magnetic-activated cell sorting (MACS; Miltenyi, Bergisch Gladbach, Germany), and subsequently cultivated in X-VIVO™ 15 serum-free hematopoietic cell medium supplemented with penicillin, streptomycin, GM-CSF (1000 IU/ml; Miltenyi, Bergisch Gladbach, Germany), and IL-4 (400 IU/ml; Miltenyi, Bergisch Gladbach, Germany) at 37 °C and 5% CO2 in a humidified atmosphere for seven days. Differentiated moDCs were matured by adding LPS (100 ng/ml; Invivogen, Toulouse, France) to the cell culture medium for 24 h and checked for cell surface marker expression using FITC anti-human CD80 (1:40 dilution, Biolegend, Cat# 305206), BV711 anti-human HLA-DR (1:100 dilution, Biolegend, Cat# 307644), and BV605 anti-human CD86 (1:400 dilution, Biolegend, Cat# 374214). The gating strategy applied for the analysis of flow cytometry-acquired data is provided in Supplementary Fig. S10. Mature moDCs were incubated with the $P_{II-1}$ peptide for 2 h prior to CD4+ T cell stimulation. CD4+ cells were isolated from PBMC of the same healthy volunteer (HV) using MACS and subsequently cultivated with penicillin, streptomycin, IL-2 (10 U/ml; R&D Systems, Minneapolis, MN, USA), and IL-7 (2.5 ng/ml; PromoKine, Heidelberg, Germany), at 37 °C and 5% CO2 in a humidified atmosphere. Cultured CD4+ cells were stimulated weekly for a total of four weeks with peptide-loaded mature moDCs and IL-12

(5 ng/ml; PromoKine, Heidelberg, Germany). The functionality of peptide-specific CD4+ T cells was analyzed by ICS.

### Antigen loading of mature moDCs

For the generation of tumor lysate, HCC cell lines transduced with the empty or the DNAJB1-PRKACA coding plasmid were treated with IFN-γ and Dox for 24 h. The treated cells were harvested, washed with PBS, subjected to five freeze–thaw cycles, irradiated with 30 Gy, and sonicated for 2 min. The clear supernatant was then added to the cell culture medium of matured moDCs of HV3, HV4, and HV5 for 24 h, and the antigen-loaded mature moDCs were subsequently harvested for HLA immunoprecipitation.

### Software and statistical analysis

The HLA allotype distribution and population coverage of the European and world population were calculated with the IEDB population coverage tool (www.iedb.org). All figures and statistical analyses were generated using GraphPad Prism 9.2.0 (GraphPad Software). *P* values of <0.05 were considered statistically significant. All flow cytometry-acquired data were analyzed with FlowJo 10.0.8 (FlowJo™ Software).

### Single-cell immune profiling

Peptide-specific CD8+ T cells of HVs induced by in vitro aAPC-based priming or in vitro amplified bulk memory CD4+ T cells of the FL-HCC patient were sorted by fluorescence-activated cell sorting (FACS), counted, and washed in 0.04% BSA/PBS according to the 10× Genomics cell preparation protocol. Single cells were partitioned into Gel Beads-in-Emulsion (GEMs) together with 10× barcoded Gel Beads and reverse transcriptase enzymatic reaction using the Chromium Controller instrument (10× Genomics, Pleasanton, CA, USA). Single-cell gene expression libraries and single-cell T-cell receptor (VDJ) libraries were then prepared using the Chromium Next GEM Single Cell 5′Kit v2 (10× Genomics, Pleasanton, CA, USA) and the Chromium Single Cell Human TCR Amplification Kit (10× Genomics, Pleasanton, CA, USA) according to the manufacturer's instructions. Libraries were pooled and sequenced on a NextSeq 550 (Illumina, San Diego, CA, USA) at 28,806, 138,975, and 3519 mean reads per cell, respectively. Samples were demultiplexed using bcl2fastq version 2.20.0.422 (Illumina, San Diego, CA, USA). Barcode processing, alignment, VDJ annotation, and single-cell 5′gene counting were performed using Cell Ranger Software version 6.0.1 (10× Genomics, Pleasanton, CA, USA). Further data processing, visualization, and analysis were done using scanpy and scirpy[75,76] for each sample separately. Cells with unique gene counts <200 and without VDJ sequence associated, as well as cells with >10% of mitochondrial genes, were removed from the analysis, keeping 474 cells (FL-HCC01), 115 cells (HV1), and 3338 cells (HV2), respectively. Data was log-normalized to a size factor of 10,000. Only highly variable genes were considered for linear dimensional reduction and were defined by a minimum mean expression of 0.0125, a maximum mean expression of 3, and a minimum dispersion of 0.5. The effect of total counts was regressed out and counts were scaled to unit variance and zero mean for each gene. The dimensionality reduction was done using principal component analysis (PCA). The neighborhood graph and UMAP embedding were computed using the rapids implementation of the UMAP algorithm[77] for 10 neighbors and the first 10 principal components (n_neighbors = 10, n_PC = 10). Unsupervised clustering was performed using the rapids implementation of the Louvain algorithm. Functional enrichment for the hallmarks of cancer gene sets[78] was performed using the decoupler's run_ora function with default parameters[79]. This analysis was performed on the log normalized counts of all genes present in more than one cell for each sample.

### Clustering of CDR3-α and CDR3-β

Clustering of variable sequences of CDR3-α and CDR3-β was conducted using GibbsCluster 2.0[80] with MHC class I configurations and a

specified core size of the smallest variable sequence in the positive dataset. The negative dataset of not binding CD3 sequences of CD8[+] T cells to the HLA class I target peptide was retrieved from the VDJdb database (https://vdjdb.cdr3.net) containing only human sequences associated with HLA-A*24:02. For the HLA class II target peptide sequences of CD4[+] T cells that did not show an in vitro response was defined as a negative dataset. The underlying position-specific scoring matrixes (PSSMs) of the clustering were used to conduct the position-wise Pearson correlation between positive and negative datasets. Correlation significance was assessed using the Pearson correlation test. Similarity and identity of variable CD3 sequences were computed using a pairwise sequence alignment by ClustalW with standard configurations (https://www.ebi.ac.uk/Tools/msa/clustalo/).

## Reporting summary

Further information on research design is available in the Nature Research Reporting Summary linked to this article.

## Data availability

The mass spectrometry proteomics data generated in this study have been deposited in the ProteomeXchange Consortium database via the PRIDE[81] partner repository under dataset identifier PXD029882. The single-cell RNA sequencing data generated in this study have been deposited in the NCBI's Gene Expression Omnibus database with the dataset identifier GSE210337. The remaining data are available within the Article, Supplementary Information or Source Data file. Source data are provided with this paper.

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

## Acknowledgements

We thank Richard Schaad, Ulrike Schmidt, Ulrich Wulle, Beate Pömmerl, and Claudia Falkenburger for technical support and the Flow Cytometry Core Facility Tübingen for cell sorting. This work was supported by the Deutsche Forschungsgemeinschaft under Germany's Excellence Strategy (Grant EXC2180-390900677 (H.-G.R., H.R.S., and J.S.W.), and CIBSS—EXC 2189—Project ID 390939984 (N.K.)), the German Cancer Consortium (DKTK) (H.-G.R., H.R.S., M. Boerries, and J.S.W.), the Deutsche Forschungsgemeinschaft (DFG, German Research Foundation, Grant WA4608/1-2 (J.S.W.); CRC/TRR167-Project ID 259373024/Z01 (M. Boerries); CRC1160-Project ID 256073931/Z02 (M. Boerries); CRC 1453-Project ID 431984000/S1 (M. Boerries); SFB1479-Project ID 441891347/S1 (M. Boerries); and SFB-1479 – Project ID: 441891347 (N.K.)), the Wilhelm Sander Stiftung (Grant 2016.177.3 (J.S.W.)), the José Carreras Leukämie-Stiftung (Grant DJCLS 05R/2017 (J.S.W.)), the Bundesministerium für Bildung und Forschung (BMBF, FKZ:01KI20130 (J.S.W.); FKZ:16LW0005 (J.S.W.); FKZ:01DP21014 (J.S.W.); FKZ 01ZZ1801B (M. Boerries)), the Else Kröner Fresenius Stiftung, Translatorik Programm (Grant 2022_EKTP03 (J.S.W.)), the Deutsche Krebshilfe (German Cancer Aid, 70114948 (J.S.W.)), the Zentren für Personalisierte Medizin (ZPM, (M. Boerries, J.S.W.)), the Applied Clinical Research program (AKF, (J.S.W.)), and the Fortüne Program of the University of Tübingen (Fortüne number 2451-0-0 (S.S.)).

## Author contributions

J.B. and J.S.W. designed the study. H.-G.R., H.R.S., S.H., M. Boerries, and N.K. provided feedback on the study design. J.B., A.N., and M.W. performed immunopeptidome experiments. J.B., Y.M., T.B., and J.R. conducted in vitro T cell experiments. M.Z., S.D., and N.K. conducted single-cell RNA sequencing. P.B., S.H., J.F., and J.B. created the DNAJB1-PRKACA fusion gene, expression model. J.B., S.D., M. Dubbelaar, and J.S. conducted data analyses. M. Denk and M.R. conducted the production of the peptide vaccine. S.S., R.K., I.B.B., J.L., U.H., M.E., I.B., J.S.H., M. Bitzer, and J.S.W. conducted patient data and sample collection as well as medical evaluation and analysis. J.B. created the figures. J.B. and J.S.W. drafted the manuscript. All authors supported the review of the manuscript.

## Funding

## Competing interests

J.B., N.K., Y.M., M.Z., S.D., M. Boerries, and J.S.W. are listed as inventors on a patent related to the DNAJB1-PRKACA T cell epitopes and TCRs (Peptides and antigen binding proteins for use in immunotherapy against fibrolamellar HCC and other cancers, Application number: EP21214728.4). H.-G.R. is listed as an inventor on a patent related to the adjuvant XS15 (Adjuvant for the induction of a cellular immune response: DE102016005550.2). The other authors declare no competing interests.

## Additional information

Jens Bauer [1,2,3], Natalie Köhler [4,5], Yacine Maringer [1,2,3], Philip Bucher [3,6], Tatjana Bilich[1,2,3], Melissa Zwick[4,7], Severin Dicks[7,8], Annika Nelde [1,2,3], Marissa Dubbelaar [1,2,3,9], Jonas Scheid [1,2,9], Marcel Wacker [1,2,3], Jonas S. Heitmann[3,10], Sarah Schroeder[1,2,11], Jonas Rieth[1,2], Monika Denk[1,2,12], Marion Richter[1,2,12], Reinhild Klein[13], Irina Bonzheim [14], Julia Luibrand[14], Ursula Holzer[6], Martin Ebinger [6], Ines B. Brecht[6], Michael Bitzer[3,12,15],

Melanie Boerries[8,16], Judith Feucht[3,6], Helmut R. Salih ●[3,10], Hans-Georg Rammensee[2,3,12], Stephan Hailfinger[3,17] & Juliane S. Walz ●[1,2,3,10,12] ✉

[1]Department of Peptide-based Immunotherapy, University and University Hospital Tübingen, Tübingen, Germany. [2]Department of Immunology, Institute for Cell Biology, University of Tübingen, Tübingen, Germany. [3]Cluster of Excellence iFIT (EXC2180) "Image-Guided and Functionally Instructed Tumor Therapies", University of Tübingen, Tübingen, Germany. [4]Department of Internal Medicine I, Medical Center - University of Freiburg, Faculty of Medicine, Albert Ludwigs University, Freiburg, Germany. [5]CIBSS – Centre for Integrative Biological Signalling Studies, University of Freiburg, Freiburg, Germany. [6]Department of Pediatric Hematology and Oncology, University Children's Hospital, University of Tübingen, Tübingen, Germany. [7]Faculty of Biology, Albert-Ludwigs-Universität, Freiburg, Germany. [8]Institute of Medical Bioinformatics and Systems Medicine, Medical Center – University of Freiburg, Faculty of Medicine, University of Freiburg, Freiburg, Germany. [9]Quantitative Biology Center (QBiC), University of Tübingen, Tübingen, Germany. [10]Clinical Collaboration Unit Translational Immunology, German Cancer Consortium (DKTK), Department of Internal Medicine, University Hospital Tübingen, Tübingen, Germany. [11]Department of Otorhinolaryngology, Head and Neck Surgery, University of Tübingen, Tübingen, Germany. [12]German Cancer Consortium (DKTK) and German Cancer Research Center (DKFZ), Partner site Tübingen, Tübingen, Germany. [13]Department of Hematology, Oncology, Clinical Immunology and Rheumatology, University Hospital Tübingen, Tübingen, Germany. [14]Department of Pathology and Neuropathology, University Hospital Tübingen, Tübingen, Germany. [15]Department of Internal Medicine I, University Hospital Tübingen, Tübingen, Germany. [16]German Cancer Consortium (DKTK), German Cancer Research Center (DKFZ) Partner Site, Freiburg, Germany. [17]Department of Medicine A, Hematology, Oncology and Pneumology, University Hospital Münster, Münster, Germany. ✉e-mail: juliane.walz@med.uni-tuebingen.de

