## [Peer Review File · Nature Communications]

The oncogenic fusion protein DNAJB1-PRKACA can be specifically targeted by peptide-based immunotherapy in fibrolamellar hepatocellular carcinomaEditorial Note: This manuscript has been previously reviewed at another journal that is not operating a transparent peer review scheme. This document only contains reviewer comments and rebuttal letters for versions considered at Nature Communications.

REVIEWERS' COMMENTS

Reviewer #1 (Remarks to the Author):

The revised manuscript by Bauer et al. has been responsive to the original critiques. The data are more clearly presented and the authors present important new data. There remain only minor aspects of the report for the authors to address.

1. Results line 70: "... promiscuous HLA binding" is not appropriate because the data at that point in the manuscript indicate use of an in silico binding prediction algorithm without demonstration of physical binding, hence "predicted" binding (at that point in the text) should be considered.

2. Given that the authors needed to use artificial APC, cytokines and a month of weekly stimulation to expand the T cells, the scRNAseq gives somewhat limited insight into the biology of the patient cells, but it does show what might be possible biology of cells that react that that peptide. The authors should be cautious when interpreting the data.

While the limited CD8+ T cell response is still mysterious, this is the state of the field, and the results presented are of keen interest to the field.

Reviewer #2 (Remarks to the Author):

The revisions improved the clarity of the manuscript.

Comment #4:

Fig.2f&g: Each dot represents a well but how many replicates were performed for each healthy individual? if "frequencies of peptide-specific CD8+ T cells compared to CD8+ T cells primed with an HLA-matched negative peptide were presented", what do the dots for the negative peptide represent? And how is this comparison performed? "% of positive" minus "% of negative"? This can be very misleading. The "raw" data (frequency of activation for the negative peptide and for PA24) should be depicted.

Reviewer #4 (Remarks to the Author):

This is an excellent paper. It comprehensively shows that the FL-HCC tumors can be targeted by neoantigen vaccines against the fusion event. Nowadays, despite the hype of neoantigens in the research fields, some have criticized that there is no hard evidence that neoantigens can be used as diagnostic/prognostic markers or vaccine targets. This work provides the exact evidence for the importance of neoantigens. I have some rather minor comments

(1) "Single-cell transcriptomics reveals TCR clonality in vaccine-induced PII-1-specific CD4+ 183 T cells" For this section, I do not seem to see any results that would be surprising or specific for this project/this patient. Almost everything seems to align with pre-existing knowledge. I suggest the authors to think logically about what these scRNA-seq data really add to their story (and clearly articulate), or otherwise tone down this section. As is written now, I am not sure I learned anything significantly new.

(2) "in particular regarding the distorted correlation between gene expression and HLA-restricted antigen presentation" I do not understand this. Could you please clarify?

(3) On a more philosophical level, if the tumor cells with this oncogenic fusion protein can grow in the patients' body (and this fusion is quite prevalent in the patient population), would it suggest that this fusion event is not very immunogenic? Then why would the neoantigen vaccine work so well, against the not-very-immunogenic neoantigens arising from this fusion event?

Point-by-point response to referees

Reviewer #1

The revised manuscript by Bauer et al. has been responsive to the original critiques. The data are more clearly presented and the authors present important new data. There remain only minor aspects of the report for the authors to address.

Author reply: We thank you very much for your kind review and highly appreciate your positive evaluation of our revised manuscript.

Please find below a detailed point to point reply to your remaining minor concerns and questions:

Comment #1: *Results line 70: "... promiscuous HLA binding" is not appropriate because the data at that point in the manuscript indicate use of an in silico binding prediction algorithm without demonstration of physical binding, hence "predicted" binding (at that point in the text) should be considered.*

Author reply: We agree with the reviewer that physical HLA binding of DNAJB1-PRKACA-derived HLA class II peptides was so far only shown for single HLA-DP and DR allotypes (Extended Data Figure 1c, Fig. 3g). Therefore, we adapted the respective results section in the revised manuscript and included the term predicted promiscuous HLA binding (line 72).

Comment #2: *Given that the authors needed to use artificial APC, cytokines and a month of weekly stimulation to expand the T cells, the scRNAseq gives somewhat limited insight into the biology of the patient cells, but it does show what might be possible biology of cells that react that that peptide. The authors should be cautious when interpreting the data.*

While the limited CD8+ T cell response is still mysterious, this is the state of the field, and the results presented are of keen interest to the field.

Author reply: We agree with the reviewer that *in vitro* expansion of T cell might alter the phenotype of peptide-specific T cells. Therefore, we added the *in vitro* expansion of peptide-specific T cells prior to characterization by flow cytometry and single cell RNA sequencing as limitation in the discussion section of the revised manuscript (lines 256-258). Of note, artificial antigen presenting cell (APC) priming was only performed for the *de novo* induction of peptide specific T cells in healthy volunteers. The vaccine patients' T cells were only *in vitro* expanded for 12 days to increase the frequency of vaccine-induced peptide-specific T cells. We adapted the material and method section in the revised manuscript to describe more clearly the usage of the different T cell expansion/priming methods (lines 487-488).

We thank the reviewer again for the positive assessment of our work and hope that the reviewer is content with revisions made and finds the manuscript suitable for publication in *Nature communications*.

Reviewer #2

The revisions improved the clarity of the manuscript.

Author reply: We thank you very much for your kind review and highly appreciate your positive evaluation of our revised manuscript.

Please find below a detailed point to point reply to your remaining concern:

Comment #1: *Fig.2f&g: Each dot represents a well but how many replicates were performed for each healthy individual? if "frequencies of peptide-specific CD8+ T cells compared to CD8+ T cells primed with an HLA-matched negative peptide were presented", what do the dots for the negative peptide represent? And how is this comparison performed? "% of positive" minus "% of negative"? This can be very misleading. The "raw" data (frequency of activation for the negative peptide and for PA24) should be depicted.*

Author reply:

We thank the reviewer for pointing the unclear description of the depicted data out. In Fig. 1f&g the "raw" data, which means the absolute frequencies of peptide-specific CD8⁺ T cells against the peptide of interest (P_{A*24} or P_{A*68/A*02}) of T cells either primed by the peptide of interest or an HLA-matched negative peptide is depicted. The frequencies were not normalized or subtracted in any way to the negative control. The term comparison does only indicate the side-by-side depiction of peptide of interest and HLA-matched negative peptide. To further improve the clarity of the Fig. 1f&g we adapted the corresponding figure legend (lines 781-786).

We thank the reviewer again for the positive assessment of our work and hope that the reviewer is content with revisions made and finds the manuscript suitable for publication in *Nature communications*.

Reviewer #4

This is an excellent paper. It comprehensively shows that the FL-HCC tumors can be targeted by neoantigen vaccines against the fusion event. Nowadays, despite the hype of neoantigens in the research fields, some have criticized that there is no hard evidence that neoantigens can be used as diagnostic/prognostic markers or vaccine targets. This work provides the exact evidence for the importance of neoantigens. I have some rather minor comments

Author reply: We thank you very much for your kind review and highly appreciate your positive evaluation of our manuscript and your input on how to further improve it. Please find below a detailed point to point reply to your specific concerns and questions:

Comment #1: *"Single-cell transcriptomics reveals TCR clonality in vaccine-induced P11-1-specific CD4+ 183 T cells" For this section, I do not seem to see any results that would be surprising or specific for this project/this patient. Almost everything seems to align with pre-existing knowledge. I suggest the authors to think logically about what these scRNA-seq data really add to their story (and clearly articulate), or otherwise tone down this section. As is written now, I am not sure I learned anything significantly new.*

Author reply:

Thank you very much for this important comment. We have performed combined single cell RNA and VDJ sequencing for the following reasons:

- To be able to discriminate between activated, exhausted and naïve T cells via their transcriptional profile and then identify their respective TCR sequences.
- To be able to generate meaningful TCR sequencing data from a low number of T cells, which was not possible by using bulk TCR sequencing.
- To validate and expand our ELISPOT and flow cytometry data with state-of-the-art technologies (as recommended by reviewer 1) using novel, more sophisticated methods.

We agree with the reviewer, that this section was not written clearly and precisely enough and have therefore re-phrased and toned down the section and integrated it into the previous paragraph. (lines 185-187 and 193-198)

Comment #2: *“in particular regarding the distorted correlation between gene expression and HLA-restricted antigen presentation” I do not understand this. Could you please clarify?*

Author reply: We thank the reviewer for turning our attention on this unclear statement. We and others have previously shown that there is only a very distorted correlation of gene expression with HLA-restricted antigen presentation which means that the immunopeptidome does neither mirror the transcriptome nor the proteome (Weinzierl *et al.* Mol Cell Proteomics 2007, Fortier *et al.* J Exp Med 2008, Berlin/Kowalewski *et al.* Leukemia 2015, Bassani-Sternberg *et al.* Mol Cell Proteomics 2015, Bassani-Sternberg *et al.* Nat Commun 2016). In particular for cancer-specific mutations it was shown that only a small fraction of alterations on DNA level result in an HLA-presented neoepitope on the cell surface that can be detected by the immune system (Yadav *et al.*, Nature 2015). Therefore, the direct analysis of HLA-presented peptides using mass-spectrometry-based immunopeptidomics seems indispensable for the identification and validation of tumor-associated antigens for immunotherapeutic approaches.

We described this point in more detail in the revised discussion section of the manuscript (lines 229-230).

Comment #3: *On a more philosophical level, if the tumor cells with this oncogenic fusion protein can grow in the patients’ body (and this fusion is quite prevalent in the patient population), would it suggest that this fusion event is not very immunogenic? Then why would the neoantigen vaccine work so well, against the not-very-immunogenic neoantigens arising from this fusion event?*

Author reply: We thank the reviewer for making this point which is a central question for the design of clinical effective cancer vaccines.

The data reported in our manuscript provide first evidence that the DNAJB1-PRKACA fusion transcript gives rise to highly immunogenic neoepitopes that can be targeted by cancer immunotherapy comprising for example peptide vaccines.

However, tumors carrying the DNAJB1-PRKACA fusion transcript, which is the main oncogenic driver in these malignancies, seem to lack/escape immune surveillance. As discussed in more detail in the author reply to comment #2 the central requirement for effective T cell-based cancer control is the natural HLA presentation of tumor antigens on the cell surface. Even if the DNAJB1-PRKACA fusion transcript is predicted to provide T cell epitopes for multiple HLA class I and class II allotypes, immunogenicity, cellular processing and HLA presentation was proven only for a selection of these peptides so far. Thus, there might be patients that, due to their specific HLA allotype or differences in the HLA processing and presenting machinery, are not able to present neoepitopes from the DNAJB1-PRKACA fusion transcript lacking neoepitope specific immune surveillance. Population effects based on specific HLA allotypes in the prevalence of DNAJB1-PRKACA carrying malignancies have, due to the rarity of this oncogenic driver, not been performed so far and require large population studies in the future.

Moreover, multiple tumors escape immune surveillance through various mechanisms including for example an immunosuppressive microenvironment at the tumor site that inhibits T cell priming to tumor antigens as well as T cell functionality.

We believe that peptide-based vaccination can overcome at least some of these escape mechanisms by enabling T cell priming and the potent expansion of tumor-specific T cells aside from the immunosuppressive microenvironment of the tumor, applying high doses of synthetic peptides

combined with adjuvants that protect peptides from immediate degradation and enable an efficient uptake and activation of antigen presenting cells at the vaccination site. Moreover, the combination of tumor-specific T cell induction by a vaccine and the general unlocking of T cell functionality using immune checkpoint inhibitor might further improve efficacy of immunotherapy.

To evaluate this concept for cancer patients carrying the DNAJB1-PRKACA fusion transcript, we are currently preparing a clinical basket study combining a neoepitope vaccine adjuvanted with the TLR1/2 agonist XS15 emulsified in Montanide ISA51VG (as successfully evaluated in Heitmann et al., *Nature* 2022 and Rammensee et al., *JIC*, 2019) with the PD-L1 antibody atezolizumab (EudraCT number 2022-002758-23). The trial will evaluate the 22-mer peptide P_{II-1} that is predicted to show promiscuous binding to multiple HLA class II allotypes and contains various peptides predicted to bind to HLA class I allotypes. Therefore, this trial will hopefully also give an answer to the first point discussed in this section i.e. which HLA allotypes give rise to DNAJB1-PRKACA neoepitopes and thus which patients will profit from a neoepitope-based vaccine.

We included a more detailed discussion on these points in the revised manuscript (line 276 to 283).

We thank the reviewer again for the positive assessment of our work and hope that the reviewer is content with revisions made and finds the manuscript suitable for publication in *Nature communications*.